# BNT162b2 SARS-CoV-2 Vaccination Elicits High Titers of Neutralizing Antibodies to Both B.1 and P.1 Variants in Previously Infected and Uninfected Subjects

**DOI:** 10.3390/life11090896

**Published:** 2021-08-29

**Authors:** Ilaria Vicenti, Francesca Gatti, Renzo Scaggiante, Adele Boccuto, Daniela Zago, Monica Basso, Filippo Dragoni, Saverio Giuseppe Parisi, Maurizio Zazzi

**Affiliations:** 1Department of Medical Biotechnologies, University of Siena, Viale Bracci, 16, 53100 Siena, Italy; vicenti@unisi.it (I.V.); boccuto2@student.unisi.it (A.B.); dragoni16@student.unisi.it (F.D.); maurizio.zazzi@unisi.it (M.Z.); 2Department of Molecular Medicine, University of Padova, Via Gabelli, 63, 35100 Padova, Italy; francesca.gatti@asst-garda.it (F.G.); daniela.zago.3@studenti.unipd.it (D.Z.); monica.basso@unipd.it (M.B.); 3Belluno Hospital, Viale Europa, 22, 32100 Belluno, Italy; renzo.scaggiante@aulss1.veneto.it

**Keywords:** SARS-CoV-2 vaccination, neutralizing antibodies, live virus neutralization, B.1 variant, P.1 variant, previously infected and uninfected subjects, health care workers

## Abstract

We aimed to investigate neutralizing antibody titers (NtAbT) to the P.1 and B.1 SARS-CoV-2 variants in a cohort of healthy health care workers (HCW), including 20 previously infected individuals tested at baseline (BL_inf_, after a median of 298 days from diagnosis) and 21 days after receiving one vaccine dose (D1_inf_) and 15 uninfected subjects tested 21 days after the second-dose vaccination (D2_uninf_). All the subjects received BNT162b2 vaccination. D1_inf_ NtAbT increased significantly with respect to BL_inf_ against both B.1 and P.1 variants, with a fold-change significantly higher for P.1. D1_inf_ NtAbT were significantly higher than D2_uninf_ NtAbT, against B.1 and P.1. NtAbT against the two strains were highly correlated. P.1 NtAbT were significantly higher than B.1 NtAbT. This difference was significant for post-vaccination sera in infected and uninfected subjects. A single-dose BNT162b2 vaccination substantially boosted the NtAb response to both variants in the previously infected subjects. NtAb titers to B.1 and P.1 lineages were highly correlated, suggesting substantial cross-neutralization. Higher titers to the P.1 than to the B.1 strain were driven by the post-vaccination titers, highlighting that cross-neutralization can be enhanced by vaccination.

## 1. Introduction

Administration of a variety of SARS-CoV-2 vaccines is proceeding at different paces in diverse geographical areas. The key question on the rate and durability of protection with the different vaccines has been further complicated by the emergence of fast-spreading variants, including the United Kingdom (B.1.1.7, alpha), South Africa (B.1.351, beta), and Brazil (P.1, gamma) lineages [1]. Indeed, SARS-CoV-2 variants escaping vaccine-induced protection have the potential to re-ignite virus transmission even in regions with optimal vaccine coverage. The first approved SARS-CoV-2 vaccines BNT162b2 and mRNA1273 have been administered to millions of people, and both are based on mRNA coding for the full-length prefusion spike glycoprotein derived from the original SARS-CoV-2 strain (lineage B). A series of small studies based on different methods for assessment of virus-neutralizing antibody (NtAb) have shown that the B.1.1.7 (alpha) variant is effectively neutralized by convalescent sera from individuals recovering from first-wave wild-type SARS-CoV-2 infection as well as by sera from BNT162b2 or mRNA1273 vaccine recipients, while neutralization of the B.1.351 (beta) variant appears to be weaker, although generally still robust [2,3,4,5,6].

Fewer and less consistent data have been obtained with the P.1 (gamma) variant. The P.1 (gamma) variant was identified for the first time in the city of Manaus, North Brazil, in December 2020 and further in passengers from Brazil tested at an airport nearby Tokyo [7]. Variant spike protein substitutions are L18F, T20N, P26S, D138Y, R190S, K417T, E484K, N501Y, D614G, H655Y, and T1027I [8]. P.1 (gamma) was defined a variant of concern because it shared mutations with other variants of concern and because of its transmissibility: a two-category dynamical model integrating genomic and mortality data estimated that P.1 (gamma) could be between 1.7 and 2.4 times more transmissible in Manaus than non-P.1 lineages [9]; moreover P.1 (gamma) led to re-infection after a previous SARS-CoV-2 infection [10]. In another cohort of individuals who received two doses of BNT162b2 or mRNA1273 vaccines, NtAb responses were also significantly decreased to the P.1 (gamma) strain, particularly in BNT162b2 vaccine recipients [2]. Interestingly, the six vaccinees reporting prior COVID-19 had high NtAb titers to most variants and exhibited substantial cross-neutralization even to the distantly related 2002–2003 SARS-CoV and pre-emergent bat-derived WIV1-CoV. This suggests that vaccination following prior infection substantially increases the breadth of cross-reactive NtAb. Similarly, Lustig et al. showed that one BNT162b2 vaccine dose increased neutralizing activity by 2 orders of magnitude against all variants tested (B.1.1.7 or alpha, B.1.351 or beta, and P.1 or gamma) in all of six previously infected health care workers (HCW) analyzed before and after vaccination [11].

## 2. Materials and Methods

To further investigate the role of BNT162b2 vaccination in NtAb titers to the P.1 variant, we studied a cohort of healthy HCW, including 20 previously infected individuals tested at baseline (BL_inf_) and 21 days after receiving one vaccine dose (D1_inf_) and 15 uninfected subjects tested 21 days after the second-dose vaccination (D2_uninf_). The previously infected group had a median age of 46 (37–52) years, included 70% females, had a laboratory diagnosis of SARS-CoV-2 infection in the Veneto region during the first outbreak in March–April 2020 when the original B.1 lineage was highly prevalent, and received a single-dose vaccination after a median of 298 (283–304) days from diagnosis. All the previously infected subjects enrolled were asymptomatic or had mild disease [12]: the diagnosis was made because of clinical suspicion or in the context of the hospital surveillance program. Eight of the twenty patients had mild disease, all with fever. A detailed description is reported in Table 1.

The uninfected group included HCW from the same region, had a median age of 49 (30–58) years with 67% females, and completed the two-dose vaccination schedule in January–February 2021. NtAb to the live virus belonging to the B.1 and P.1 lineages, as assessed by NGS (submitted to GISAID, accession number EPI_ISL_2472918 for P.1 and EPI_ISL_2472896 for B.1), were titrated in duplicate by testing two-fold serial dilutions of sera with 100 TCID_50_ of the corresponding SARS-CoV-2 strain in Vero E6 cells with automated measurement of cell viability by the Cell-titer Glo 2.0 system in a GloMax Discover luciferase plate reader (Promega, Madison, WI, USA) [13,14,15]. The NtAb titer (ID_50_) was defined as the reciprocal value of the sample dilution that showed a 50% protection from the virus-induced cytopathic effect. Each run included an uninfected control, an infected control, and the virus back-titration to confirm the virus inoculum. NtAb titers were expressed as median (IQR), and the non-parametric Wilcoxon signed-rank sum test and Mann–Whitney test were used to analyze changes in paired and unpaired data, respectively. Spearman analysis was used to measure the correlation between NtAb titers to the two viral strains. Analyses were run in IBM SPSS Statistics, version 20 (IBM Corp., Armonk, NY, USA) and all *p*-values were 2-sided.

Written informed consent was obtained from all the HCW to be enrolled in a neutralizing antibody (NtAb) follow-up study, as approved by the comitato per la sperimentazione clinica di Treviso e Belluno (protocol code 812/2020).

## 3. Results

In the previously infected patient group, D1_inf_ NtAb titers increased significantly with respect to BL_inf_ against both the B.1 (median values 1706 (993–3186) vs. 38 (16–67); *p* < 0.001) and the P.1 (median values 4087 (3324–5053) vs. 66 (30–129); *p* < 0.001) variant, with a fold-change significantly higher for the P.1 variant vs. the B.1 wild type (94 (29–134) vs. 44 (15–70); *p* = 0.03).

At 20 days post-vaccination, D1_inf_ NtAb titers were significantly higher than D2_uninf_ NtAb titers against both the B.1 (median values 1706 (993–3186) vs. 190 (119–302); *p* < 0.0001) and the P.1 (median values 4087 (3324–5053) vs. 288 (147–904) *p* < 0.0001) variant. Overall, NtAb titers against the two strains were highly correlated (rho = 0.924, *p* < 0.001). However, P.1 NtAb titers were significantly higher with respect to B.1 NtAb titers (median 394 (73–368) vs. 225 (75–1227), *p* < 0.001). Interestingly, this difference was significant for post-vaccination data in infected and uninfected subjects (D1_inf_: 4087 (3227–5125) vs. 1706 (993–3186), *p* < 0.001; D2_uninf_ 288 (147–904) vs. 190 (119–302), *p* = 0.02) but not for BL_inf_ (66 (29–130) vs. 38 (16–167), *p* = 0.13) (Figure 1).

Outcomes of studies focusing on P.1 and BNT162b2 are summarized in Table 2.

## 4. Discussion

The NtAb response to SARS-CoV-2 vaccination is being increasingly investigated to determine the impact of virus variability on vaccine efficacy. The P.1 variant is of particular concern due to the resurgence of high infection rates in Brazilian areas with supposedly large prevalence of first-wave infection [26]. Various previous studies have shown that vaccination of uninfected subjects with B lineage mRNA vaccines elicits lower but measurable NtAb titers to the P.1 variant compared to the other variants [2]. In our group of 15 uninfected BNT162b2 vaccinees, the NtAb response to the P.1 variant was not decreased, or even higher, compared to the original B.1 lineage virus. Notably, in the 20 previously infected subjects included in our study, single-dose vaccination substantially boosted the NtAb response to both variants tested, supporting the data reported by Lustig et al. in six analogous patients whose sera had high NtAb titers to the B.1.1.7, B.1.351, and P.1 lineages [11]. This may support the recommendation for vaccinating previously infected patients [27]. However, the increased titer and breadth of NtAb could simply reflect the expected secondary immune response without conferring stronger immunity to secondary infection.

The NtAb titer after a single vaccine dose in previously infected subjects was higher that NtAb titer in uninfected HCW after a complete vaccination schedule: this result is in accord with data published by Mazzoni et al. [28], who had uninfected subjects tested after a shorter (7 days after the second dose) and a longer (50 days from the first vaccine dose) interval than ours (21 days after the second dose). Conversely, comparable titers were reported in the study by Gobbi et al. [29] and in that by Ebinger et al. [30]: possible explanations for the different results may be the low numerosity (15 subjects) for the first one and the racial distribution for the second one (white people were about 50%).

The more intense humoral response in SARS-CoV-2-exposed subjects seemed not related to the vaccine type administered: the anti-receptor-binding domain of SARS-CoV-2 spike protein antibody titers was significantly higher in HCWs with a previous natural infection 4 weeks after the second dose of inactivated SARS-CoV-2 vaccine (CoronaVac) [31]. Of note, we should take into account the risk of an excess of immune activation, followed by switch-off of the response for antigen exhaustion: this mechanism may explain the reduced antibody response after the second vaccine dose in subjects with a previous infection [28,32]. Finally, the extreme heterogeneity of the methods used in the different studies, commercial or in-house methods, using recombinant viruses or authentic viral strains, as in our case, often make this evidence poorly comparable.

NtAb titers to B.1 and P.1 lineages were highly correlated in the whole case file. Intriguingly, paired analysis revealed higher titers to the P.1 than to the B.1 strain. This overall effect was driven by the post-vaccination titers, highlighting that. Interestingly, as recently published, patients infected with the B.1.351.V2 South African variant had higher NtAb titers to the related B.1.351.V3 variant than to the infecting B.1.351.V2 strain [33] and uninfected recipients of the BNT162b2 vaccine responded better to a recombinant virus mimicking the B.1.1.7 variant than to the vaccine strain [6]. These data show that SARS-CoV-2 variants may be well controlled following natural or artificial immunization with a different variant [34]. The characteristics of the antigenicity of the P.1 variant are not completely known [35]: this fact and the different study designs, viral components tested (strain isolated from a clinical sample or generated mutant, as in Chang et al. [36]), and commercial or in-house methods used to evaluate different kinds of antibody titers may explain the multifaceted response to the P.1 variant with respect to other SARS-CoV-2 strains. Gidari et al. [17] described a post-vaccinal NtAb titer to B1 and P.1 specular to ours, and sera from vaccinated individuals showed a significant loss of neutralizing activity against P.1 but the impact was lower than that against B.1.351 [24,25,37]: these data underline the need to include SARS-CoV-2 viral sequencing of PCR-positive COVID-19 subjects in the diagnostic workflow because variant diffusion may reduce vaccine efficacy.

The strength of our work lies in the homogeneity of the cohort, in the history of mild or asymptomatic infection, in the long interval from infection to pre-vaccine evaluation, and in the use of two authentic viral isolates into a classical neutralization assay. It must be noted that a number of factors can complicate the interpretation of NtAb data, including ethnicity, use of different test virus strains, and the NtAb titration technology, particularly the live virus vs. pseudovirus or spike recombinant format.

While most studies appear to support a clinically valuable breadth of the NtAb response irrespective of the infecting or vaccine virus variant, surveillance as well as development of reference isolates and technical standards remain mandatory to manage the impact of ongoing SARS-CoV-2 variability on the control of infection.

## Figures and Tables

**Figure 1 life-11-00896-f001:**
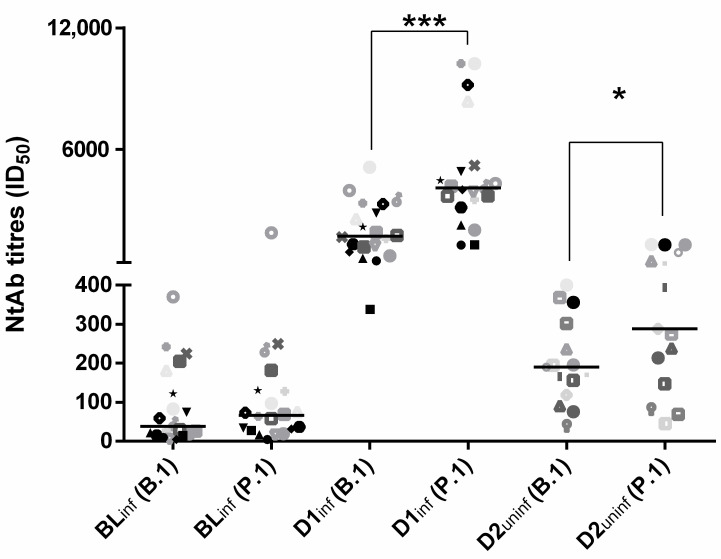
Neutralizing antibody titers to the original B.1 lineage and the P.1 SARS-CoV-2 variant in 20 infected subjects tested at baseline (BL_inf_) and 21 days following single-dose BNT162b2 vaccination (D1_inf_) and in 15 uninfected subjects tested 21 days following second-dose BNT162b2 vaccination (D2_uninf_). Data are reported as individual ID_50_ values and as median value at each study time. The same symbols indicate the same subjects at different time points. Asterisks indicate significance levels: *, *p* < 0.05, ***, *p* < 0.001.

**Table 1 life-11-00896-t001:** Description of signs and/or symptoms of the health care workers with mild COVID-19.

Patients	Age (Years)	Fever	Cough	Asthenia
Pt 1	46	x	x	
Pt 2	48	x	x	x
Pt 3	32	x	x	x
Pt 4	47	x		
Pt 5	49	x	x	
Pt 6	54	x	x	
Pt 7	45	x	x	
Pt 8	44	x		

**Table 2 life-11-00896-t002:** Description of the studies focusing on the neutralizing antibody response in subjects vaccinated with BNT162b2 mRNA COVID-19 vaccine.

N	Authors and Year of Publication	Study Population	Country	Total Number of Subjects Enrolled	History of Natural SARS-CoV-2 Infection	Study Points	Neutralizating Antibody Method	SARS-CoV-2 Strains Tested Other Than P.1	Main Results
1	Zani et al., 2021 [16]	Volunteers	Italy	37	No	Between 10 and 20 days after administration of second dose of vaccine	Cytopathic-effect-based assay using authentic viruses isolated in Italy	B.1, B.1.1.7, B.1.351, B.1.525	All the serum samples efficiently neutralized the SARS-CoV-2 B.1 lineage and all the viral variants. As compared with neutralization of the SARS-CoV-2 B.1 lineage, neutralization of the SARS-CoV-2 P.1 lineage was robust but significantly lower.
2	Gidari et al., 2021 [17]	Vaccinated health care workers, candidates as hyper-immune plasma donors, patients with ascertained SARS-CoV-2 P.1 infection	Italy	202	Yes (*n* = 112) and no (*n* = 90)	Median of 16 days after the second dose in vaccinated health care workers, median of 67 days after diagnosis of infection in candidates as hyper-immune plasma donors, median of 21 days after diagnosis of P.1 infection	In-house microneutralization assay	20A.EU1, B.1.1.7	B.1.1.7 and P.1 were less efficiently neutralized by convalescent wild-type infected serums if compared to the 20A.EU1 strain. BNT162b2 vaccine-elicited human sera showed equivalent neutralization potency against the B.1.1.7 variant, but it was significantly lower against the P.1 variant. Convalescent P.1 patients showed an important reduction in neutralizing antibodies against 20A.EU1 and B.1.1.7.
3	Barros-Martins et al., 2021 [18]	Health care professionals vaccinated with first dose of AstraZeneca’s ChAdOx1-nCov-19 (ChAd) and with second dose of the same vaccine or of BNT162b2	Germany	87	No	Before and 3 weeks after booster with ChAd or with BNT162b2	Enzyme-linked-immunosorbent-assay-based surrogate virus neutralization test	Wuhan, B.1.1.7, B.1.351	Both vaccines boosted prime-induced immunity, and BNT162b2 induced high titers of neutralizing antibodies against the B.1.1.7, B.1.351, and P.1 variants.
4	Lustig et al., 2021 [19]	Healthy health care workers	Israel	36	No	1 month following receipt of second vaccine dose	Microneutralization assays	Sub-lineage B.1 (hCoV-19/Israel/CVL-45526-ngs/2020), alpha (hCoV-19/Israel/CVL-46879-ngs/2020), beta (hCoV-19/Israel/CVL-2557-ngs/2020), and delta sample 1 (S1, hCoV-19/Israel/CVL-12804/2021 and S2, hCoV-19/Israel/CVL-12806/2021)	There was significant fold-change reduction in neutralizing titres compared with the original virus for gamma (P.1), beta (B.1.351), and delta variants. The reduction in the alpha (B.1.1.7) variant was not significant.
5	Collier et al., 2021 [20]	Community participants or health care workers	United Kingdom	140	Yes (*n* = 10) and no (*n* = 130)	3 to 12 weeks after first of dose vaccine and again 3 weeks after second dose of vaccine	Pseudotyped virus neutralization assays	Wild type, B.1.1.7, B.1.351	Sera from participants above 80 showed lower neutralization potency against the B.1.1.7, B.1.351, and P.1. variants of concern (VOC) than against the wild-type virus and were more likely to lack any neutralization against VOC following the first dose. However, following the second dose, neutralization against VOC was detectable regardless of age.
6	Stankov et al., 2021 [21]	Health care professionals	Germany	231	Yes (*n* = 83) and no (*n* = 148)	Mean of 17.6 days after first dose and a mean of 21 days after second dose	cPass Neutralization Antibody Detection kit (GenScript)	SARS-CoV-2 wild type, B1.1.7 B.1.351	A single vaccine dose may frequently fail to induce a measurable neutralizing antibody response.
7	Leier et al., 2021 [22]	Participants	United States	30	Yes (*n* = 10), and no (*n* = 20)	Immediately after receiving first dose of vaccine and at least 14 days after second vaccine dose	Focus reduction neutralization test	B.1.1.7, B.1.351, USA257 WA1/2020	Neutralizing antibody titers increased in previously infected vaccinees relative to uninfected vaccinees against every variant tested: B.1.1.7, B.1.351, P.1, and original SARS-CoV-2.
8	Anichini et al., 2021 [23]	Health care workers	Italy	60	No	30 days after second dose of vaccine	Microneutralization assay	Native Wuhan, B.1.1.7, B.1.351	The neutralizing antibody titers elicited against the wild-type strain showed a slight decrease versus the P1 lineage and a significant decrease to the B.1.351 lineage. No significant differences were found in comparison with the B.1.1.7 lineage.
9	Caniels et al., 2021 [24]	Convalescent COVID-19 patients and vaccinated health care workers	Netherlands	119	Yes (*n* = 69) and no (*n* = 50)	Sera of SARS-CoV-2-infected adults collected 4 to 6 weeks after symptom onset, sera collected 4r weeks after second dose of vaccine in 50 health care workers	Pseudovirus neutralization assay, authentic virus neutralization assay	Wild type, B.1.1.7, B.1.351	Substantial neutralizing activity was present against the WT virus in 96% of convalescent patients irrespective of hospitalization and in all vaccine recipients. There was a marked and significant reduction in serum ability to neutralize pseudovirus variants of concern. For all three groups, the difference was most apparent against B.1.351.
10	Dejnirattisai et al., 2021 [25]	Participants	United Kingdom	59	Yes (*n* = 34) and no (*n* = 25)	Serum collected 4–14 days following second dose of vaccine, samples collected 4-9 weeks following infection in previously infected	Live virus neutralization test	Victoria (SARS-CoV-2/human/AUS/VIC01/2020), B.1.1.7, B.1.351	In convalescent samples, P.1 mean neutralization titres were reduced compared to Victoria. This reduction was similar to B.1.1.7 and considerably less than B.1.351. In vaccine serum, mean neutralization titres against P.1 were reduced to the Victoria virus. Neutralization titres against P.1 were similar to those against B.1.1.7, with a significant reduction against B.1.351.
11	Garcia-Beltran et al., 2021 [2]	Participants	USA	30	No (except one suspected case)	Serum collected 7–32 days following second vaccine dose	Lentivirus-vector pseudovirus neutralization test	Wild type, B.1.1.7, B.1.1.298, B.1.429, P.2, B.1.351	There was a decrease in neutralization relative to the wild type for B.1.1.7, for B.1.1.298, for B.1.429, for P.1, and for B.1.351.

## Data Availability

Raw data can be made available by the corresponding author upon reasonable request.

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
