# Peer review of "BNT162b2 SARS-CoV-2 Vaccination Elicits High Titers of Neutralizing Antibodies to Both B.1 and P.1 Variants in Previously Infected and Uninfected Subjects"

_life, 2021, doi:10.3390/life11090896_

Round 1
Reviewer 1 Report
The authors obtained interesting results when comparing the immune response to the vaccination in individuals previously infected with SARS-CoV-2 versus non-infected individuals. Differences were also observed between the different strains of the virus. However, some formal aspects should be corrected prior to publication. Furthermore, other aspects related to the experimental design or the results interpretation should be clarified.
The experimental design is clear in the materials and methods section, but not in the abstract. Furthermore, it is better not show the numerical data in the abstract, but only the main results and conclusions.
It would be convenient to describe in a table the characteristics, including the clinical, of the population studied.
The populations should have the same number of individuals without previous infection than the previously infected (20) to be balanced. Since the n is small, a 25% difference could be important. In addition, it should be easier to obtain samples from previously uninfected vaccinated individuals, since they are the majority.
The conclusion that affirms that "cross-neutralization can be enhanced by vaccination, is speculative". As the authors point "NtAb titers to B.1 and P.1 lineages were highly correlated, SUGGESTING substantial cross-neutralization" (there are other possible explanations for this fact). It is not appropriate to speak of cross-neutralization for the same specie.
The introduction should indicate which are the strains that escape from the vaccine. The authors speak about the different protection against strains B.1.17 and B.1.351, but the study determine the protection for all variants of the strain B1 together. Since the virus samples are sequenced, the different strains within B1 should be indicated.
Statistical analysis could be performed separately for men and women, since the percentage of women is higher . Besides, different responses to infection have been observed between the different sexes.
It is not appropriate to speak about homologous viruses for the same species.
The authors should discusse the results obteined in Lucas et al, 2021 https://doi.org/10.1101/2021.07.14.21260307 , showing that plasma from previously infected vaccinated individuals displayed better neutralization compared to plasma from uninfected individuals who received two vaccine doses. These authors find differences between the different strains of the virus.
Also the works by Gobbi et al, 2021 (https://doi.org/10.3390/v13030422); Soysal et al, 2021 (https://doi.org/10.1080/21645515.2021.1953344); Ebinger et al, 2021 (https://doi.org/10.1038.s41591-021-01325-6) ; Mazzoni et al, 2021 (https://doi.org/10.1172/JCI149150) ; Levi et al, 2021 (https://doi.org/10.1172/JCI149154) ; should be discussed, since are closely related, and sometimes coincident, or obtain opposite results.
Reviewer 2 Report
The report is well-written, the methods are soundings. The findings are not entirely new, given a plenty of references that the authors have not cited and that should be added to discussion. A Table summarizing outcomes of studies about P.1 and BNT162b2 so far would be of great interest to the reader. Ref 8 discusses outcomes after a single dose, which is of low value given the current schedules. Ref.7 points to mRNA-1273 : if the authors intend to discuss the efficacy of different vaccines, then it would be an entirely different article type, so I suggest focusing on BNT162b2 only and removing the reference to mRNA-1273..
- Mohsen M, Bachmann MF, Vogel M, Augusto GS, Liu X, Chang X. BNT162b2 mRNA COVID-19 vaccine induces antibodies of broader cross-reactivity than natural infection but recognition of mutant viruses is up to 10-fold reduced. 2021:2021.03.13.435222. doi: 10.1101/2021.03.13.435222 %J bioRxiv.
- Wang P, Wang M, Yu J, Cerutti G, Nair MS, Huang Y, et al. Increased Resistance of SARS-CoV-2 Variant P.1 to Antibody Neutralization. 2021:2021.03.01.433466. doi: 10.1101/2021.03.01.433466 %J bioRxiv.
- Dejnirattisai W, Zhou D, Supasa P, Liu C, Mentzer AJ, Ginn H, et al. Antibody evasion by the Brazilian P.1 strain of SARS-CoV-2. 2021:2021.03.12.435194. doi: 10.1101/2021.03.12.435194 %J bioRxiv
- Gidari A, Sabbatini S, Bastianelli S, Pierucci S, Busti C, Monari C, et al. Cross-neutralization of SARS-CoV-2 B.1.1.7 and P.1 variants in vaccinated, convalescent and P.1 infected. J Infect. 2021. doi: 10.1016/j.jinf.2021.07.019.
- Caniels TG, Bontjer I, van der Straten K, Poniman M, Burger JA, Appelman B, et al. Emerging SARS-CoV-2 variants of concern evade humoral immune responses from infection and vaccination. 2021:2021.05.26.21257441. doi: 10.1101/2021.05.26.21257441 %J medRxiv.
- Imai M, Halfmann PJ, Yamayoshi S, Iwatsuki-Horimoto K, Chiba S, Watanabe T, et al. Characterization of a new SARS-CoV-2 variant that emerged in Brazil. Proceedings of the National Academy of Sciences of the United States of America. 2021;118(27). doi: 10.1073/pnas.2106535118
